# HumanSplat: Generalizable Single-Image Human Gaussian Splatting with Structure Priors

**Panwang Pan**[*1], **Zhuo Su**[*†1], **Chenguo Lin**[*1,2], **Zhen Fan**[1], **Yongjie Zhang**[1], **Zeming Li**[1],
**Tingting Shen**[3], **Yadong Mu**[2], **Yebin Liu**[‡4]
[*] Equal contribution     [†] Project lead     [‡] Corresponding author

[1]ByteDance, [2]Peking University, [3]Xiamen University, [4]Tsinghua University

## Abstract

Despite recent advancements in high-fidelity human reconstruction techniques, the requirements for densely captured images or time-consuming per-instance optimization significantly hinder their applications in broader scenarios. To tackle these issues, we present **HumanSplat**, which predicts the 3D Gaussian Splatting properties of any human from a single input image in a generalizable manner. Specifically, HumanSplat comprises a 2D multi-view diffusion model and a latent reconstruction Transformer with human structure priors that adeptly integrate geometric priors and semantic features within a unified framework. A hierarchical loss that incorporates human semantic information is devised to achieve high-fidelity texture modeling and impose stronger constraints on the estimated multiple views. Comprehensive experiments on standard benchmarks and in-the-wild images demonstrate that HumanSplat surpasses existing state-of-the-art methods in achieving photorealistic novel-view synthesis. Project page: https://humansplat.github.io.

## 1 Introduction

Realistic 3D human reconstruction is a fundamental task in computer vision with widespread applications in numerous fields, including social media, gaming, e-commerce, telepresence, etc. Previous works for single-image human reconstruction can be broadly categorized into explicit and implicit approaches. Explicit methods [1, 2, 3, 4], such as those based on parametric body models like SMPL [5, 6], estimate human meshes by directly optimizing parameters and clothing offset to fit the observed image. However, these methods often struggle with complex clothing styles and require lengthy optimization. Implicit methods [7, 8, 9, 10, 11] represent humans using continuous functions, such as occupancy [7], SDF [12], and NeRF [13]. While they excel at modeling flexible topology, these methods are limited in scalability and efficiency due to the high computational cost associated with training and inference. Recent advances in 3D Gaussian Splatting (3DGS) [14] have provided a balance between efficiency and rendering quality for reconstructing detailed 3D human models, which however rely on multi-view image [15, 16, 17] or monocular video [18, 19, 20] input. Recent popular human reconstruction studies [21, 22, 23, 24] focus on the score distillation sampling (SDS) technique [25] to lift 2D diffusion priors to 3D, but time-consuming optimization (e.g., 2 hours) is required for each instance. Some generalizable and large-reconstruction-model-based works [10, 26, 27, 28] can directly generalize the regression of 3D representations but either disregard human priors or require multi-view inputs, limiting the stability and feasibility in downstream applications.

In this paper, we propose a novel generalizable approach **HumanSplat** for single-image human reconstruction by introducing a generalizable Gaussian Splatting framework with a fine-tuned 2D multi-view diffusion model and well-conceived 3D human structure priors. Different from existing human 3DGS methods, our approach directly infers Gaussian properties from a single input image,

38th Conference on Neural Information Processing Systems (NeurIPS 2024).

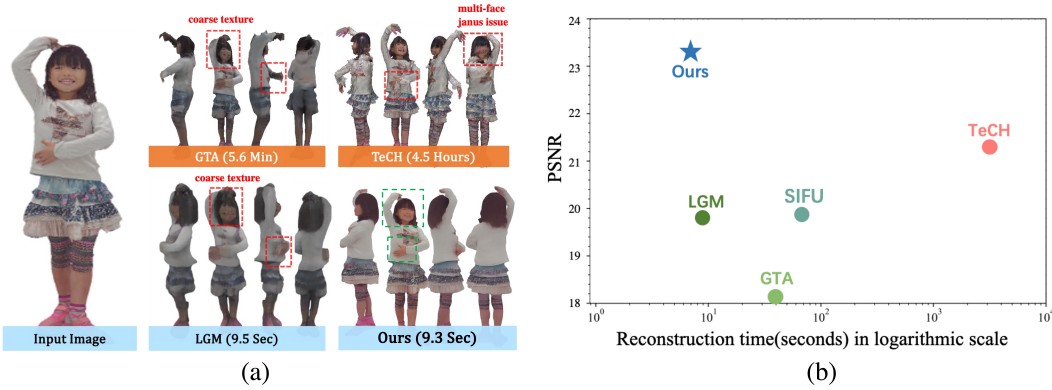

Figure 1: Our method achieves state-of-the-art rendering quality while maintaining the fastest runtime. (a) Qualitative results: LGM [29] and GTA [30] are generalizable but in lower quality, TeCH [21] exhibits issues with multi-face rendering and is time-consuming. In contrast, our method achieves higher fidelity in a much shorter time. (b) Performance and runtime comparison: metrics are evaluated on the challenging Twindom dataset.

eliminating the requirement for per-instance optimization or densely captured images. This empowers our method to generalize effectively in diverse scenarios while delivering high-quality reconstructions.

The key insight behind our method is to reconstruct Gaussian properties from the diffusion latent space in a generalizable end-to-end architecture, integrating a 2D generative diffusion model as appearance prior and a human parametric model as structure prior. Specifically, facing this under-constraint reconstruction problem from single-view input with extensive invisible regions, we first leverage a 2D multi-view diffusion model (denoted as **novel-view synthesizer**) to hallucinate the unseen parts of clothed humans. Then a generalizable **latent reconstruction Transformer** is proposed to enable interaction among the generated diffusion latent embeddings and the geometric information of the structured human model, enhancing the quality of 3DGS reconstruction. During interactions, to mitigate the limitations of inaccurate human priors like the SMPL model, we devise a projection strategy to strike a balance between robustness and flexibility, projecting 3D tokens into the 2D space and conducting searches within adjacent windows using projection-aware attention. Another specific challenge of human reconstruction is to capture fine detail in visually sensitive areas, such as the face and hand. To address this problem, we exploit the semantic cues from the structure priors and propose **semantics-guided objectives** to further promote the fine details reconstruction quality. By synergistically amalgamating these crucial components into a unified framework, our method achieves state-of-the-art performance in striking the right balance between quality and efficiency.

In summary, the main contributions of our work are as follows:

- We propose a novel generalizable human Gaussian Splatting network for high-fidelity human reconstruction from a single image. To the best of our knowledge, it is the first to leverage latent Gaussian reconstruction with a 2D generative diffusion model in an end-to-end framework for efficient and accurate single-image human reconstruction.

- We integrate structure and appearance cues within a universal Transformer framework by leveraging human geometry priors from the SMPL model and human appearance priors from the 2D generative diffusion model. Geometric priors stabilize the generation of high-quality human geometry, while appearance prior helps hallucinate unseen parts of clothed humans.

- We enhance the fidelity of reconstructed human models by introducing semantic cues, hierarchical supervision, and tailored loss functions. Extensive experiments demonstrate that our method achieves state-of-the-art performance, surpassing existing methods.

## 2   Related Work

**Single-Image Human Reconstruction.**   Single-image human reconstruction methods fall into explicit and implicit approaches. Explicit methods use parametric body models [5, 6] to estimate

minimally clothed human meshes [1, 2, 31, 32, 33, 34, 35, 36, 37], adding clothing via 3D offsets [3, 4, 38, 39, 40, 41] or garment templates [42, 43, 44]. However, these methods are constrained by topology, particularly with loose clothing. Implicit methods [7, 8, 9, 10, 11, 22, 27, 45, 46], utilizing representations like occupancy, signed distance fields (SDF) and Neural Radiance Fields (NeRF), offer flexibility with topology, allowing for accurate depiction of 3D clothed humans. Recent methods [30, 47, 48, 49, 50, 51, 52, 53, 54, 55, 56, 57, 58, 59, 60, 61] combine implicit representation with explicit human priors for better robustness. Among these, GTA [30] uses Transformers with fixed embeddings for translating image features into 3D tri-plane features. TeCH [62] employs diffusion-based models for visualizing unseen areas but needs extensive optimization and accurate SMPL-X models. Another approach, HumanSGD [22], uses diffusion models for texture inpainting but struggles with mesh inaccuracies from [8]. NeRF-based methods like SHERF [10] generate human NeRF from single images using hierarchical features, while ELICIT [11] leverages CLIP [63] for contextual understanding. These NeRF-based methods produce high-quality images but often struggle with detailed 3D mesh generation and require extensive optimization time.

**Human Gaussian Splatting.** While implicit representations like SDF or NeRF struggle with balance optimization efficiency and rendering quality, the high efficiency of 3DGS [14] has advanced 3D human creation. Recent 3DGS methods have targeted multi-view videos, monocular video sequences, and sparse-view input. Multi-view video methods like D3GA [15], HuGS [16], and 3DGS-Avatar [17] exploit dense spatial and temporal information for detailed models. Additionally, HiFi4G [64] combine 3DGS with a dual-graph mechanism to preserve spatial-temporal consistency, ASH [65] utilizes mesh UV parameterization for real-time rendering, and Animatable Gaussians [66] improve garment dynamics via pose projection mechanism and 2D CNNs. The monocular video only provides sufficient observation as humans move, so the monocular human Gaussian Splatting methods [18, 19, 20, 67, 68, 69, 70, 71, 72, 73] often resort to using LBS for mapping to a canonical space and introducing additional regularization terms. For sparse-view images, GPS-Gaussian [28] achieves high rendering speeds without per-subject optimization by using a feed-forward way with efficient 2D CNNs encoding from diverse 3D human scan data. In comparison, our work stands out as the first one-shot generalizable human GS method, using only single image input and achieving high-quality reconstruction without any optimization and fine-tuning.

**Generalizable Large Reconstrucion Model.** Large reconstruction model (LRM) [74, 75] presents that with sufficient parameters and training datasets, a deterministic feed-forward Transformer [76] is capable of learning a triplane feature for volumetric rendering from a single-view image. Targeting general 3D object, TriplaneGaussian [77] combines 3DGS with triplane as a hybrid 3D representation for fast rendering, where point clouds are first extracted from an image and then projected on triplane features to decode 3DGS attributes. Most recent methods further leverage 2D diffusion models that can generate multi-view images simultaneously [78, 79, 80, 81] to provide plausible and abundant inputs for the following reconstruction, so they focus on the sparse multi-view reconstruction task. Instant3D [81] follows the technique of LRM and produces triplane features from 4 posed images by a Transformer encoder. CRM [82], MeshLRM [83] and InstantMesh [84] replaces triplane NeRF by FlexiCubes [85] to extract meshes directly. LGM [29], GRM [86] and GS-LRM [87] predict 3DGS attributes from each input pixel [88, 89] and combine the Gaussians from multiple viewpoints as the final 3D output. For human task, CharacterGen [90] follows Instant3D [81] to adopt a two-stage approach: first, generating a 2D multi-view canonical images, then using multi-view SDF reconstruction for cartoon digital characters, without direct interaction between stages. A more compact and elegant method HumanLRM [27] replaces training data of LRM [74] from general objects with human data to predict triplane NeRF, showcasing the capacities of the large model. While it relies heavily on the dataset, often leading to issues like missing limbs due to a lack of prior knowledge. In contrast, our 3DGS-based method leverages the latent diffusion and reconstruction model in an end-to-end framework and introduces human priors by addressing their inaccuracy with specific projection strategies, achieving better generalization and stability with even less data.

## 3 Method

### 3.1 Preliminary

**SMPL** [5] is a widely-used parametric human body model that is created by skinning and blend shapes, and learned from thousands of 3D body scans. The SMPL model [5] utilizes shape parameters

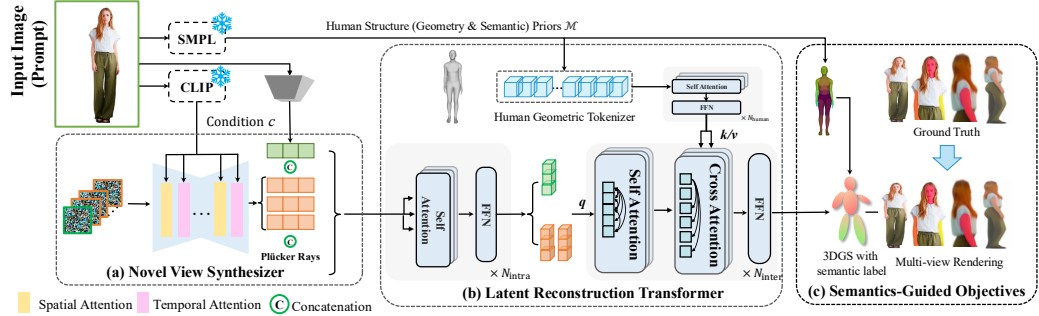

Figure 2: Overview of HumanSplat. (a) Multi-view latent features are first generated by a fine-tuned multi-view diffusion model (Novel View Synthesizer in Sec. 3.3). (b) Then, the Latent Reconstruction Transformer (Sec. 3.4) interacts global latent features (Sec. 3.4.1) and human geometric prior (Sec. 3.4.2). (c) Finally, the semantic-guided objectives (Sec. 3.5) are proposed to reconstruct the final human 3DGS.

$\boldsymbol{\beta} \in \mathbb{R}^{10}$ and pose parameters $\boldsymbol{\theta} \in \mathbb{R}^{24\times3}$ to parameterize the deformation of the human body mesh $\mathcal{M}(\boldsymbol{\beta}, \boldsymbol{\theta}) : \boldsymbol{\beta} \times \boldsymbol{\theta} \mapsto \mathbb{R}^{6890\times3}$.

**3D Gaussians Splatting** [14] is a recently proposed 3D representation that formulates 3D content by a set of $\mathbf{N_p}$ colored Gaussians $\mathcal{G} = \{\mathbf{g}_i\}_{i=1}^{\mathbf{N_p}}$. Each Gaussian $\mathbf{g}_i = \exp\left(-\frac{1}{2}(\mathbf{x} - \boldsymbol{\mu}_i)^\top \boldsymbol{\Sigma}_i^{-1}(\mathbf{x} - \boldsymbol{\mu}_i)\right)$ has opacity $\boldsymbol{\sigma}_i \in \mathbb{R}$ and color $\mathbf{c}_i \in \mathbb{R}^3$ attributes, where $\boldsymbol{\mu}_i \in \mathbb{R}^3$ is its location and $\boldsymbol{\Sigma}_i \in \mathbb{R}^{3\times3}$ implies its scaling $\mathbf{s_i} \in \mathbb{R}^3$ and orientation quaternion $\mathbf{q_i} \in \mathbb{R}^4$ in 3D space. These attributes constitute $\mathbf{G} \in \mathbb{R}^{\mathbf{N_p}\times14}$ can represent a 3D object, and by projecting to the image plane with opacity-based color composition in the depth order, it can be differentiably rendered into 2D images in real-time.

## 3.2 Overview

Given a single input image of a human body $\mathbf{I_0} \in \mathbb{R}^{H\times W\times3}$, our task is to reconstruct a 3D representation, i.e., 3DGS in this work, from which novel-view images can be rendered. As illustrated in Fig. 2, our method leverages a 2D diffusion model and a novel latent reconstruction Transformer that adeptly integrates 2D appearance priors, human geometric priors, and semantic cues within a universal framework. First, we use SMPL estimator [91, 92] to predict human structure priors $\mathcal{M}$, and CLIP [63] to generate the image embedding of the input image $\mathbf{c}$. A latent temporal-spatial diffusion model [93], referred to as the **novel-view synthesizer** (Sec. 3.3), is fine-tuned and produces multi-view latent features $\left\{\mathbf{F}_i \in \mathbb{R}^{h\times w\times c}\right\}_{i=1}^{\mathbf{N}}$, where $\mathbf{N}$ is the number of views, and $h$, $w$ and $c$ are the height, width and channel of the latent features. Then, we utilize a novel **latent reconstruction Transformer** (Sec. 3.4) that adeptly integrates human geometric prior and latent features within a universal framework, which predicts Gaussian attributes $\mathbf{G} \coloneqq \left\{(\boldsymbol{\mu}_i, \boldsymbol{q}_i, \boldsymbol{s}_i, \boldsymbol{c}_i, \boldsymbol{\sigma}_i)|i = 1, ..., \mathbf{N_p}\right\}$ and rendered into new images, where $\mathbf{N_p}$ represents the number of Gaussian points. Furthermore, to achieve high-fidelity texture modeling and better constrain the estimated multiple views, we designed a hierarchical loss that incorporates human semantic priors (See 3.5). Note that during training, the network is trained using 3D scan data to ensure accurate supervision from various viewpoints, where the registered SMPL is fitted via the multi-view version of SMPLify [92]. During inference, the model employs PIXIE [91] to predict human structure priors $\mathcal{M}$ and synthesize novel views from only a single image input based on the trained model, without any fine-tuning and optimization.

## 3.3 Video Diffusion Model as Novel-view Synthesizer

We take advantage of the pre-trained video diffusion model SV3D [93] as appearance prior. For the input image $\mathbf{I}_0$, we leverage a CLIP image encoder [63] to obtain the image embedding $\mathbf{c}$ and a pre-trained VAE [94] $\mathcal{E}$ to acquire latent feature $\mathbf{F}_0$ as conditions. Subsequently, we progressively denoises gaussian noises into temporal-continuous $\mathbf{N}$ multi-view latent features $\{\mathbf{F}_i\}_{i=1}^{\mathbf{N}}$ by a spatial-temporal UNet $D_\theta$ [95] with the objective:

$$\mathbb{E}_{\epsilon\sim p(\epsilon)} \left[\lambda(\epsilon)\|D_\theta(\{\mathbf{F}_i^\epsilon\}_{i=1}^{\mathbf{N}}; \{e_i, a_i\}_{i=1}^{\mathbf{N}}, \mathbf{c}, \mathbf{F}_0, \epsilon) - \{\mathbf{F}_i\}_{i=1}^{\mathbf{N}}\|_2^2\right], \tag{1}$$

where $\{\mathbf{F}_i^{\epsilon}\}_{i=1}^{\mathbf{N}}$ is the noise-corrupted multi-view latent features, $\{e_i, a_i\}_{i=1}^{\mathbf{N}}$ is the corresponding relative elevation and azimuth angles to $\mathbf{I}_0$, $p(\epsilon)$ is the noise probability distribution and $\lambda(\epsilon) \in \mathbb{R}^+$ is the loss weighting term related to the noise level $\epsilon$. Since SV3D is merely pre-trained on the dataset of objects [96], we further fine-tune it using human datasets to improve its effectiveness in the human reconstruction task. More technical details are available in Appendix A.1.

## 3.4 Latent Reconstruction Transformer

Latent reconstruction Transformer contains two parts: **latent embedding interaction** (Sec. 3.4.1) and **geometry-aware interaction** (Sec. 3.4.2), which seamlessly incorporate latent features from novel-view synthesizer and human structure priors into the reconstruction process. The two components together form a cohesive framework for accurately recovering intricate human details.

### 3.4.1 Latent Embedding Interaction

We utilize a pretrained VAE [94] encoder $\mathcal{E}$ to encode the input image $\mathbf{I}_0$ into a latent representation $\mathbf{F}_0 = \mathcal{E}(\mathbf{I}_0) \in \mathbb{R}^{h \times w \times c}$ and combine it with the generated $\{\mathbf{F}_i\}_{i=1}^{\mathbf{N}}$ (Sec. 3.3). Following previous works [97, 98], latent representations are concatenated with their corresponding Plücker embeddings [99] along the channel dimension, resulting in a dense pose-conditioned feature map. These latents are then divided into non-overlapping patches [100] and mapped to $d$-dimensional latent tokens by a linear layer. An intra-attention module is followed to thoroughly extract spatial correlations of latent tokens, as shown in Fig. 3, by a standard Transformer [76] with $N_{\text{intra}}$ blocks of multi-head self-attention and feed-forward network (FFN):

$$\bar{\mathbf{F}}_i = [\text{FFN}(\text{SelfAttention}(\mathbf{F}_i))]_{\times N_{\text{intra}}}. \tag{2}$$

### 3.4.2 Geometry-aware Interaction

**Human Geometric Tokenizer.** Given a human geometric model $\mathcal{M} \in \mathbb{R}^{6890 \times 3}$, to obtain the feature representation, we project $\mathcal{M}$ onto the input view and compute the feature vector through bilinear interpolation on the feature grids by

$$\mathbf{\Pi}_0(\mathcal{M}) = \mathbf{K} \left( \mathbf{R}_0 \mathcal{M} + \mathbf{t}_0 \right), \tag{3}$$

where $\mathbf{R}_0$, $\mathbf{t}_0$ are camera extrinsic parameters of the input view, $\mathbf{K}$ is the intrinsic parameters. After the projection operation, we obtain geometric human prior tokens by concatenating $\mathcal{M}$ and $\mathbf{F}_0[\mathbf{\Pi}_0(\mathcal{M})]$, and mapping them to $d$ dimension. $\mathbf{F}_0[\mathbf{\Pi}_0(\mathcal{M})]$ stands for querying the input latent feature $\mathbf{F}_0$ from projected locations. Similar to Latent Embedding Interaction (Sec. 3.4.1), an intra-attention module is utilized to interact among these human prior tokens and get human geometric features $\bar{\mathbf{H}} \in \mathbb{R}^{6890 \times d}$.

**Human Geometry-aware Attention.** Empirically, we find that incorporating human priors [5, 6, 101] (e.g., SMPL model) into human reconstruction faces a significant dilemma of the trade-off between robustness and flexibility [30, 102]. On the one hand, the SMPL model provides geometric priors that alleviate common issues (e.g., broken body parts and various artifacts). On the other hand, it struggles to accurately capture and represent a broad spectrum of complex clothing styles, particularly more fluid garments like dresses and skirts. This limitation highlights a fundamental shortcoming in the model's capacity to accurately depict diverse apparel. As shown in Fig. 3, HumanSplat addresses inaccuracies in human priors by projecting 3D tokens into 2D space and conducting local searches within adjacent windows, efficiently using priors while minimizing redundancy. Specifically, we introduce projection-aware attention within the inter-attention module, utilizing a window $\mathbf{W}(K_{\text{win}} \times K_{\text{win}})$ and employing masked multi-head cross-attention. Here, the latent features $\{\bar{\mathbf{F}}_i\}_{i=1}^{\mathbf{N}}$ serve as queries, and the human prior tokens $\bar{\mathbf{H}}$ serve as keys and values. The process can be formulated as follows:

$$\{\tilde{\mathbf{F}}_i\}_{i=1}^{\mathbf{N}} = [\text{FFN}(\text{CrossAttention}_{\text{mask}}(\{\bar{\mathbf{F}}_i\}_{i=1}^{\mathbf{N}}, \bar{\mathbf{H}}))]_{\times N_{\text{inter}}}, \tag{4}$$

where feature interactions occur when $\mathbf{\Pi}(\bar{\mathbf{H}}, \mathbf{K}, \mathbf{R}_i, \mathbf{t}_i)$ is within the window of $\bar{\mathbf{F}}_i$. It is noteworthy that the complexity of the cross-attention operation is reduced from $\mathcal{O}(\mathbf{L}_F \times \mathbf{L}_H)$ to $\mathcal{O}(\mathbf{L}_F \times K_{\text{win}}^2)$, compared to the vanilla multi-head cross-attention setting, where $\mathbf{L}_F$ and $\mathbf{L}_H$ represent the lengths of the latent and human geometric tokens respectively.

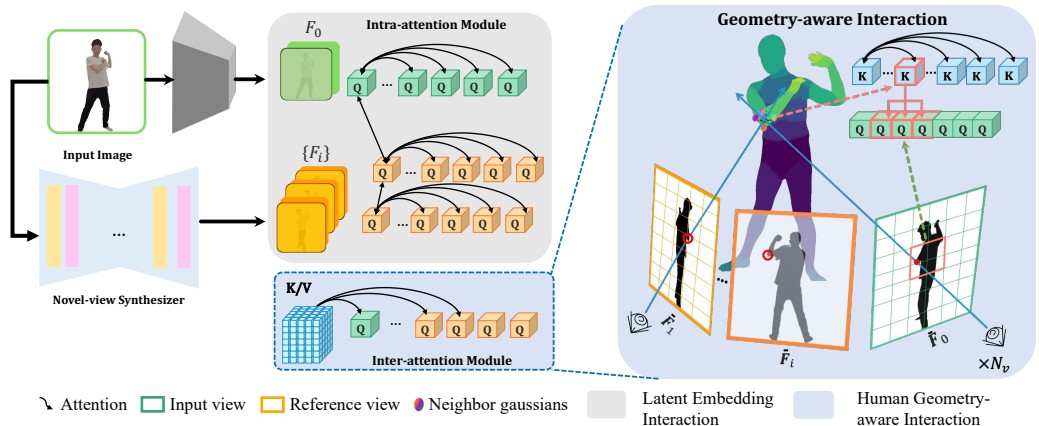

Figure 3: Illustration of latent reconstruction Transformer. It first divides $\mathbf{F}_0$ and $\mathbf{F}_i$ into non-overlapping patches, which are then processed through an intra-attention module (Sec. 3.4.1). Within the iter-attention module (Sec. 3.4.2), we introduce the projection-aware attention with a window $\mathbf{W}(K_{\text{win}} \times K_{\text{win}})$, and the attributes of 3D Gaussians are decoded with a Conv $1 \times 1$ layer.

## 3.5 Semantics-guided Objectives

From each output token $\tilde{\mathbf{F}}_i$, we decode the attributes of pixel-aligned Gaussians $\mathbf{G}$ in the corresponding patch by a convolutional layer with a $1 \times 1$ kernel. These attributes are used to render novel-view images through 3D Gaussian Splatting rasterization. An ideal training objective should ensure that the rendered outputs closely match the supervised images, striving for overall consistency between the reconstructed renderings and the ground truth. However, human attention tends to be particularly focused on facial regions, where intricate details play a pivotal role in perception. Therefore, to enhance the fidelity of reconstructed human models, we optimize our objective functions to preferentially focus on the facial regions.

**Hierarchical Loss.** Traditional approaches often overlook the semantic richness inherent in human anatomy, leading to reconstructions that lack crucial details and accuracy. To address this, we propose a novel framework that harnesses semantic cues, hierarchical supervision, and tailored loss functions to guide the training process. This ensures that not only are the overall structures of the human body accurately represented, but also that critical areas such as the face are reconstructed with exceptional detail and fidelity. Specifically, by incorporating a human prior model equipped with semantic information, which adheres to the widely-used definition of 24 human body parts, such as "right hand", "left hand", "head", etc. The ground-truth body part segmentation is obtained from SMPL meshes with predefined semantic vertices [103]. Therefore, we can utilize different attention weights for different body parts and render different levels of real images from different viewpoints to provide hierarchical supervision. This approach significantly facilitates the precise localization of essential body parts, including the head, hands and arms. Please refer to Appendix A.2 for more detailed procedural insights. The overall loss is defined as a weighted sum of part-specific losses:

$$\mathcal{L}_H = \frac{1}{n}\frac{1}{m}\sum_{i=1}^{n}\sum_{j=1}^{m} \lambda_i \lambda_j (\mathcal{L}_{\text{MSE}}(\mathbf{I}_{i,j}, \hat{\mathbf{I}}_{i,j}) + \lambda_p \mathcal{L}_{\text{p}}(\mathbf{I}_{i,j}, \hat{\mathbf{I}}_{i,j})), \tag{5}$$

where $i \in \{1, ..., n\}$ and $j \in \{1, ..., m\}$ denote different resolution levels and human parts. $\hat{\mathbf{I}}_{i,j}$ and $\mathbf{I}_{i,j}$ are the predicted and ground-truth data for part $j$, $\lambda_i$ and $\lambda_j$ are weighting factors that reflect the relative importance of each resolution level and each body part, respectively. $\mathcal{L}_{\text{MSE}}$ measures the MSE loss and $\mathcal{L}_p$ measures the perceptual loss [104].

**Reconstruction Loss.** Reconstruction loss $\mathcal{L}_{\text{Rec}}$ based on the $\mathbf{N}_{\text{render}}$ rendered multi-view images is defined as

$$\mathcal{L}_{\text{Rec}} = \sum_{i=1}^{\mathbf{N}_{\text{render}}} \mathcal{L}_{\text{MSE}}(\mathbf{I}_i, \hat{\mathbf{I}}_i) + \lambda_m \mathcal{L}_{\text{MSE}}(\mathbf{M}, \hat{\mathbf{M}}_i) + \lambda_p \mathcal{L}_{\text{p}}(\mathbf{I}_i, \hat{\mathbf{I}}_i), \tag{6}$$

Table 1: Quantitative comparison on texture against other methods. * indicates that this method is fine-tuned on our training dataset. † indicates that this method requires per-instance optimization.

| Method | Category | | THuman2.0 [105] | | | Twindom [106] | | | Time |
|---|---|---|---|---|---|---|---|---|---|
| | Diffusion | Human Prior | PSNR↑ | SSIM↑ | LPIPS↓ | PSNR↑ | SSIM↑ | LPIPS↓ | |
| PIFu [107] | ✗ | ✗ | 18.093 | 0.911 | 0.137 | - | - | - | 30.0s |
| LGM* [29] | ✓ | ✗ | 20.013 | 0.893 | 0.116 | 19.840 | 0.851 | 0.292 | 9.5s |
| GTA [30] | ✗ | ✓ | 18.050 | - | - | 17.669 | 0.741 | 0.418 | 43s |
| SIFU [102] | ✗ | ✓ | 22.025 | 0.921 | 0.084 | 19.714 | 0.832 | 0.312 | 44s |
| SIFU† [102] | ✓ | ✓ | 22.102 | 0.923 | 0.079 | - | - | - | 6min |
| Magic123† [108] | ✓ | ✗ | 14.501 | 0.874 | 0.145 | - | - | - | 1h |
| HumanSGD† [22] | ✓ | ✓ | 17.365 | 0.895 | 0.130 | - | - | - | 7min |
| TeCH† [21] | ✓ | ✓ | **25.211** | **0.936** | 0.083 | 21.192 | 0.884 | 0.188 | 4.5h |
| **HumanSplat** (Ours) | ✓ | ✓ | 24.033 | 0.918 | **0.055** | **23.346** | **0.913** | **0.125** | **9.3s** |

where $\mathbf{I}_i$ and $\hat{\mathbf{I}}_i$ are the ground-truth images and render images via Gaussian Splatting, $\mathbf{M}_i$ and $\hat{\mathbf{M}}_i$ are original and rendered foreground mask. $\lambda_m$ and $\lambda_p$ are hyperparameters for balancing three kinds of losses. The final training objective is a combination of the two loss terms: $\mathcal{L} = \mathcal{L}_H + \mathcal{L}_{\text{Rec}}$.

# 4 Experiments

## 4.1 Implementation Details

**Training Details.** HumanSplat is trained on 500 THuman2.0 [105], 1500 2K2K [109], and 1500 Twindom [106] high-fidelity human scans. We evenly position 36 cameras across each of three hierarchical cycles to capture the full body, half body, and face, with rendering resolution set to $512 \times 512$. We conduct 200 epochs of 256-res training with a learning rate of 1e-5 and a batch size of 32 over 2 days on 8 A100 (40G VRAM) GPUs, while 512-res finetuning costs 2 additional days. We train our model with AdamW [110] optimizer, whose $\beta_1$, $\beta_2$ are set to 0.9 and 0.95 respectively. A weight decay of 0.05 is used on all parameters except those of the LayerNorm layers. We use a cosine learning rate decay scheduler with a 2000-step linear warm-up and the peak learning rate is set to 4e-4. For parts related to the head, hands, and arms, $\lambda_j$ are set to 2, while the rest human part are set to 1. The parameters $\lambda_i$, $\lambda_p$ and $\lambda_m$ are set to 1. The model is trained for 80K iterations on 256-res and then fine-tuned on 512-res for another 20K iterations. To enable efficient training and inference, we adopt Flash-Attention-v2 [111] in the xFormers [112] library, gradient checkpointing [113], and FP16 mixed-precision [114]. Fine-tuning the novel view synthesizer takes about 18 hours. Please refer to Appendix A.1 for more details about novel view synthesizer fine-tuning.

**Inference Time.** The video diffusion model takes about 9s to generate multi-view latent features while the subsequent latent reconstruction for 3DGS takes only about 0.3s. Additionally, it can render novel views at a rate exceeding 150 FPS on a NVIDIA A100 GPU.

## 4.2 Comparison

**Quantitative Comparison.** We conduct a quantitative comparison on 21 THuman2.0 [105] and 51 Twindom [106] scans, rendering textured meshes from multiple angles and using PSNR, SSIM [115], and VGG-LPIPS [104] as evaluation metrics. To ensure fairness, we fine-tune LGM [29] on our datasets and fully optimize each TeCH [21] instance for comparison. As shown in Tab. 1, HumanSplat consistently outperforms previous generalizable methods across all datasets. Specifically, HumanSplat surpasses SIFU [102] by +1.92 (8.72%) in PSNR and reduces LPIPS from 0.079 to 0.055. Notably, HumanSplat also excels on the more challenging Twindom dataset, outperforming TeCH [21] by +2.15 (10.16%) in PSNR and reducing LPIPS from 0.188 to 0.125. Although the values can further decrease with model training, we control the convergence to the current extent in the dataset for better generalization under more general in-the-wild scenarios. Besides, we also report the reconstruction times for $512 \times 512$ input images on a NVIDIA A100 GPU. HumanSplat achieves a remarkably fast reconstruction time, approximately 9.3 seconds on a single NVIDIA A100 GPU, making it much more practical than TeCH's 4.5-hour reconstruction time.

**Qualitative Comparison.** As illustrated in Fig. 4 and Fig. 5, by incorporating semantics-guided objectives, Humansplat achieves more detailed and higher fidelity results against GTA [30] and

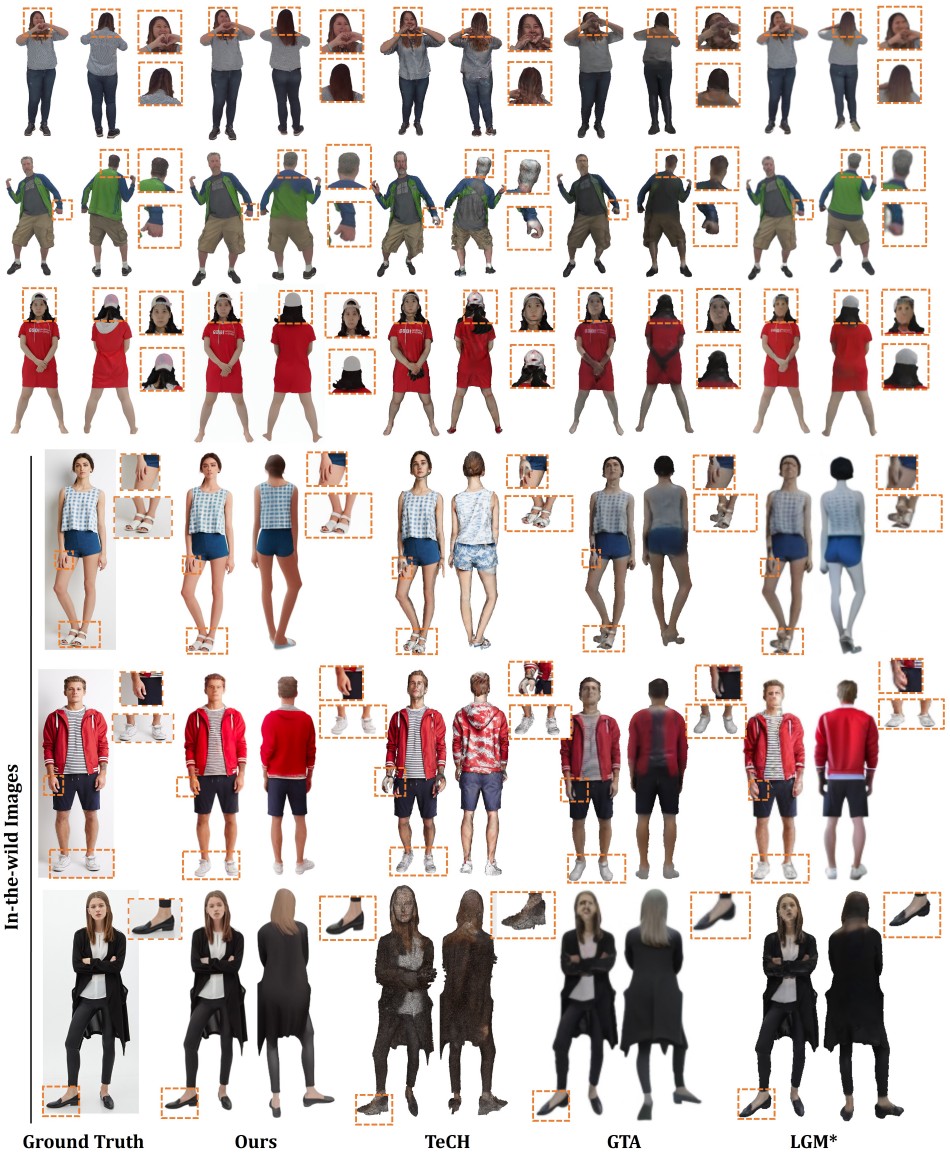

| Ground Truth | Ours | TeCH | GTA | LGM* |

Figure 4: Qualitative comparison of ours against TeCH [21], GTA [30] and LGM [29] on Thuman2.0 [105], Twindom [106] and in-the-wild images. Our method achieves the highest quality. Note that TeCH achieves clearer results but fails to preserve the face identity.

LGM [29]. Although TeCH [21] also generates high-quality images, its SDS optimization often results in overly saturated multi-face outcomes [116, 117]. Moreover, while TeCH delivers the highest PSNR on the Thuman2.0 dataset, the LPIPS metric in Tab. 1, confirms that our method provides more realistic results. As shown in Fig. 6, Humansplat outperforms HumanSGD on in-the-wild images from Adobe Stock, effectively predicting 3D Gaussian Splatting properties without per-instance optimization. Notably, the results on in-the-wild images further demonstrate our strong generalization capability and our model provides superior textures on both the front and invisible regions, highlighting the effectiveness of incorporating appearance and geometric priors.

### 4.3 Ablation Study

**Latent Reconstruction Transformer.** HumanSplat leverages the Stable Diffusion latent space for feature extraction. Experimental results demonstrate that utilizing the compact latent space for reconstructions leads to better rendering quality compared to using the pixel space, as detailed in

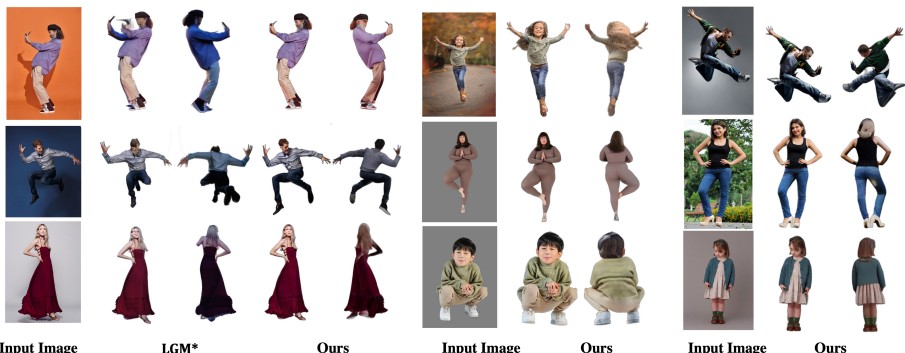

Figure 5: Qualitative results showcasing reconstructions of humans in challenging poses, diverse identities, and varying camera viewpoints from in-the-wild images.

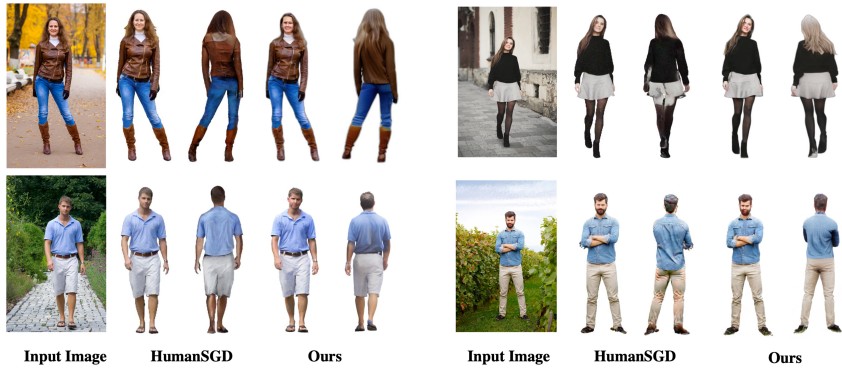

Figure 6: Qualitative Comparison of ours against HumanSGD [22] on in-the-wild images.

Tab. 2 (a). We also find that patching in the time dimension for latent decoding by temporal VAE [95] adversely affects the quality, so we employ the vanilla VAE [94] decoder in this work.

**Human Geometric Prior.** Tab. 2 (b) illustrates the significance of human geometric prior in our approach. By utilizing SMPL as the geometric prior for human representation, we assess the robustness of HumanSplat with respect to different SMPL parameters. We report both the estimated [2] and Ground Truth SMPL parameters [92] and compare the model that does not incorporate a human structure prior. These experiments provide insight into the benefit of incorporating a human structure prior, as significant performance degradation is observed without it. Furthermore, thanks to treat human priors as keys and values, HumanSplat exhibits robustness in testing against challenging poses with imperfect estimation, with only minor declines in metrics. The effectiveness of the human geometric prior is also demonstrated qualitatively in Fig. 7 (a).

**Window Size.** As analyzed in Sec. 3.4.2, human priors are crucial for the robustness of the human reconstruction model. As evidenced in Tab. 2 (c), when the window size $K_{\text{win}}$ is set to 1 (dubbed "Projection Only"), employing human priors significantly enhances performance, improving the PSNR metric from 22.635 to 23.333. We further explore setting window size to $K_{\text{win}} = 2$ in a latent space with a resolution of $64 \times 64$, which further improves the model's PSNR metric from 23.333 to 24.294. This improvement is attributed to Humansplat with $K_{\text{win}} = 2$ is capable of handling loose clothing and tolerating imprecise SMPL parameters. In Tab. 3, we present results for the 2K2K dataset across various window sizes. Further refinement with a window size of $K_{\text{win}} = 3$ continues to enhance the results, whereas using $K_{\text{win}} = 4$ does not yield additional improvements.

**Semantics-guided Training Objectives.** Thanks to the incorporation of semantic-guided hierarchical losses, HumanSplat can generate higher-quality textures for both the face and hand, while also promoting the convergence of the training process, as depicted in Fig. 7 (a) and (b). Furthermore, Fig. 7 (c) demonstrates that HumanSplat further produces 3D coherent segmentation results, which have potential applications in editing and specific generation of individual parts of 3D humans.

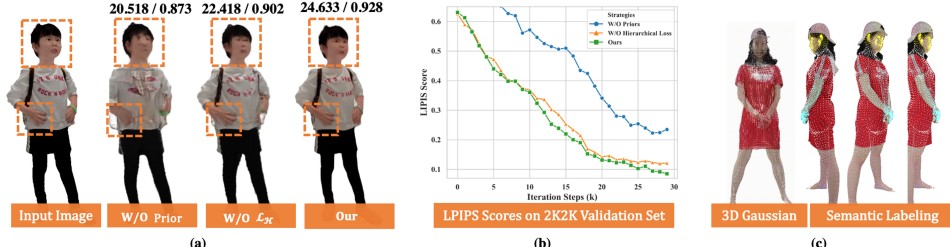

Figure 7: (a) Qualitative evaluation of human prior and hierarchical loss $\mathcal{L}_{\mathcal{H}}$. PSNR/SSIM values are shown at the top of each image, presenting the effectiveness of the two strategies in the full pipeline. (b) LPIPS scores during training on the 2K2K validation set. (c) HumanSplat can provide 3D coherent segmentation results for potential applications.

Table 2: Ablation study of several proposed designs on 2K2K [109] evaluation dataset.

(a) Reconstruction Space.

| Space | PSNR↑ | SSIM↑ | LPIPS↓ |
|---|---|---|---|
| Pixel Space | 23.125 | 0.892 | 0.066 |
| Temporal VAE [95] | 24.133 | 0.902 | 0.047 |
| **Ours** [94] | **24.293** | **0.915** | **0.040** |

(b) Geometric Prior Initialization.

| Method | PSNR↑ | SSIM↑ | LPIPS↓ |
|---|---|---|---|
| GT SMPL | **24.633** | **0.935** | **0.025** |
| w/o SMPL | 22.635 | 0.893 | 0.182 |
| **Ours** [2] | 24.293 | 0.915 | 0.040 |

(c) Different projection methods.

| Method | PSNR↑ | SSIM↑ | LPIPS↓ |
|---|---|---|---|
| w/o Human Prior | 22.635 | 0.893 | 0.182 |
| Projection Only | 23.333 | 0.916 | 0.057 |
| **Ours** ($K_{\text{win}} = 2$) | **24.293** | **0.915** | **0.040** |

Table 3: Evaluation of different window sizes $K_{\text{win}}$ on the 2K2K [109] evaluation dataset.

| Method | PSNR ↑ | SSIM ↑ | LPIPS ↓ |
|---|---|---|---|
| w/o Human Prior | 22.635 (-1.658) | 0.893 (-0.022) | 0.182 (+0.142) |
| $K_{\text{win}} = 1$ | 23.333 (-0.960) | 0.916 (-0.001) | 0.057 (+0.017) |
| $K_{\text{win}} = 3$ | 24.383 (+0.089) | 0.931 (+0.016) | 0.041 (+0.001) |
| $K_{\text{win}} = 4$ | 24.013 (-0.280) | 0.922 (+0.007) | 0.048 (+0.008) |
| $K_{\text{win}} = 2$ **(Ours)** | 24.293 | 0.915 | 0.040 |

## 5 Conclusion

We present HumanSplat, a pioneering generalizable human reconstruction network that derives 3D Gaussian Splatting properties from a single image. This model integrates generative diffusion and latent reconstruction Transformer models with human structure priors, enhanced by a tailored semantic-aware hierarchical loss. These innovations achieve high-fidelity reconstruction results in a feed-forward manner without any optimization or fine-tuning, especially in the important focal areas such as the face and hands. Extensive experiments demonstrate that HumanSplat surpasses existing state-of-the-art methods in both quality and efficiency, including robust performance in handling challenging poses and loose clothing. This capability opens up a broad spectrum of potential applications.

**Limitations and Future Works.** Our method has some limitations that can be addressed in future works: (a). While effective with most attire, the model may struggle with intricate garments and accessories. Future enhancements should incorporate 2D data and innovative training strategies, which is a promising direction for improving the handling of diverse and unusual clothing styles. (b). Although already efficient, increasing the computational speed by constraining Gaussian's number or combining compact 3DGS with texture representation could further facilitate real-time applications, particularly on mobile devices. (c). Animating the reconstructed human models currently requires post-processing. Future work could introduce canonical space to reconstruct animatable avatars directly, streamlining the process and improving efficiency.

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

# A Implementation Details

## A.1 Additional Novel View Synthesizer Details

**Conditions of Diffusion Model.** SVD and SV3D [93, 95] replace CLIP text embeddings [63] with the image embedding of the conditioning. Compared to text prompts, image prompts provide information more accurately and directly, and also save time that would otherwise be spent on text inversion or image caption [21, 102]. SV3D takes an image and the relative camera orbit elevation and azimuth angles as inputs to generate several corresponding novel-view images, which are continuous in 3D space and can be regarded as a video with the camera moving. In our work, we finetuned SV3D on human datasets with relative camera elevation and azimuth angles to generate 4 orthogonal views, which we found to be sufficient for reconstruction [102].

**Transformer Block.** The CLIP embeddings obtained from the conditioning image is fed into the cross-attention layers of each Transformer block, acting as keys and values, while the feature at that layer serves as the queries. Moreover, the camera trajectory and the diffusion noise timestep are incorporated into the residual blocks.

**Fine-tuning Details.** We involves the widely used EDM [118, 119] framework, incorporating a simplified diffusion loss for finetuning. We fine-tune the novel-view synthesizer at 512-res uses 1000-step warmup with the peak learning rate 1e-4 in the cosine learning rate decay schedule. We use a per-GPU batch size of 8 objects during 256-res training, and a per-GPU batch size of 4 during 512-res finetuning stage. For each instance, we use 4 input views and 4 novel supervision views at each iteration of 256-res training and 512-res finetuning.

**Triangular CFG Scaling.** **C**lassifier-**f**ree **G**uidance (**CFG**) [120] is a widely used technique to trade off controllability with diversity. However, this scaling causes the last few frames in our generated orbits to be over-sharpened. To tackle this issue, we use a triangle wave CFG [93, 121] scaling during inference: linearly increase CFG from 1 at the front view to 2.5 at the back view, then linearly decrease it back to 1 .

**3D Consistency Evaluation.** We also conduct an evaluation of the 3D consistency of our novel-view synthesizer using the advanced local correspondence matching algorithm, MASt3R [122]. This approach involves one-to-one image matching between input views and their generated novel views, with the average number of matching correspondences serving as our metric. Our synthesizer exhibits a notable improvement in 3D consistency for human novel-view generation after fine-tuning with human datasets. Specifically, the original SV3D achieves an average of 723.25 matching points, while our novel-view synthesizer increases this number to 930.13. In comparison, the ground truth demonstrates a matching point number of 1106.33. This evaluation highlights that fine-tuning on human datasets significantly enhances multi-view consistency metrics.

## A.2 Additional Latent Reconstruction Transformer Details

**Network Architecture.** The network architecture of the proposed latent reconstruction Transformer is shown in Fig. 8. The framework is composed of a latent encoder with intra- and inter-attention and a Gaussian parameter prediction module. It takes initial multi-view latent embeddings as input and performs projection-aware attention to aggregate human geometric information. From each output token, we decode the attributes of pixel-aligned Gaussians in the corresponding patch with a Conv $1 \times 1$ layer.

**Hierarchical Loss.** Our approach is efficient and can be utilized online without pre-computation. We calculate the visible face triangles given the mesh and camera parameters. Then each visible triangular face is assigned the corresponding semantic label [103]. Additionally, it provides labels with superior 3D consistency and avoids issues associated with training instability. First, the hierarchical loss is specifically focused on the head and hands, excluding hair and clothing, which can be further supervised using reconstruction loss $\mathcal{L}_{\text{Rec}}$. Second, instead of using segmentation to reconstruct and then concatenate different components, we design our loss function based on part segmentation. Thank to this design, HumanSplat is tolerant of inaccuracies in segmentation.

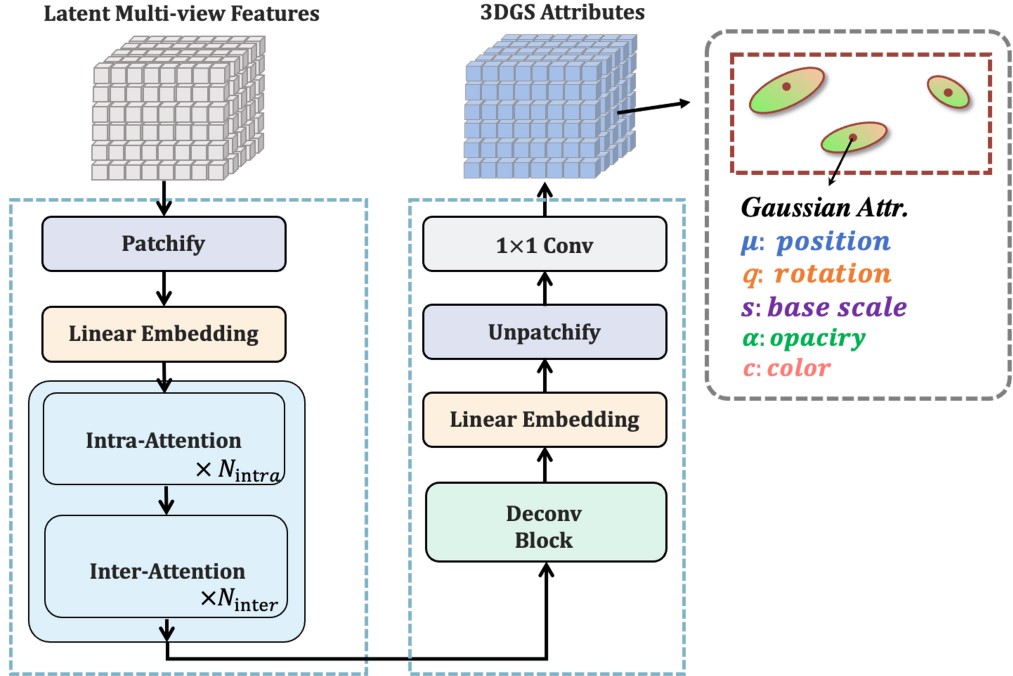

Figure 8: Detailed network architecture of latent reconstruction Transformer.

## B More Results

For practical applications demanding high input view reconstruction quality, which could potentially be directly derived from the input image. We improve the weights of the reconstruction loss for input views (referred to as **reweighting loss**), demonstrating that the current model with reweighting loss exhibits sufficient capacity to render input views with higher fidelity, as illustrated in Fig.9. We also conduct an ablation experiment for the reweighting loss function on the 2K2K Dataset [109]. There are slight declines in quantitative metrics, specifically, HumanSplat with reweighting loss achieves a numerical change of -0.141, -0.015, and +0.006 in PNSR, SSIM, and LPIPS. This suggests that reweighting loss is not "a bag of freebies" and HumanSplat without reweighting loss strikes a balance between the quality of novel views and the input views.

As for novel view and novel pose rendering, more results are shown in Fig. 10 and Fig. 11. For a single input image, we use Open-AnimateAnyone [123] for novel pose synthesis and HumanSplat for novel view synthesis. This functionality enables dynamic timestamps and real-time rendering of novel views, thereby enhancing immersive virtual exploration. Please also kindly refer to our supplementary video for more results.

## C Broader Impacts

While our model demonstrates the capability to reconstruct photo-realistic 3D avatars, it also introduces potential risks such as privacy violations. To mitigate these concerns, it is imperative to establish robust ethical guidelines and legal frameworks. This necessitates a collaborative effort among researchers, developers, and policymakers to promote the responsible use of this technology and safeguard against its potential misuse.

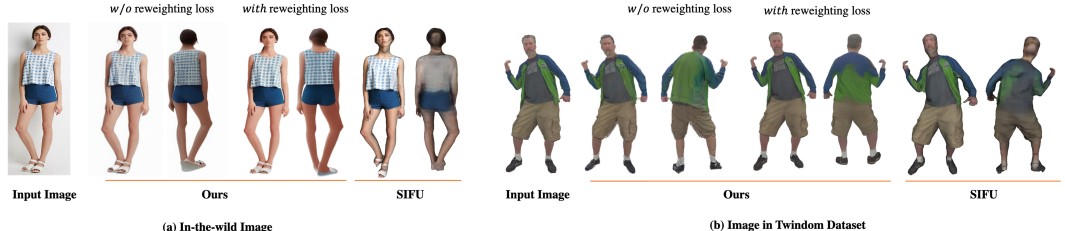

Figure 9: **Ablation Study on Reweighting Loss.** Comparison of the original HumanSplat, trained with reweighting loss, and SIFU featuring texture.

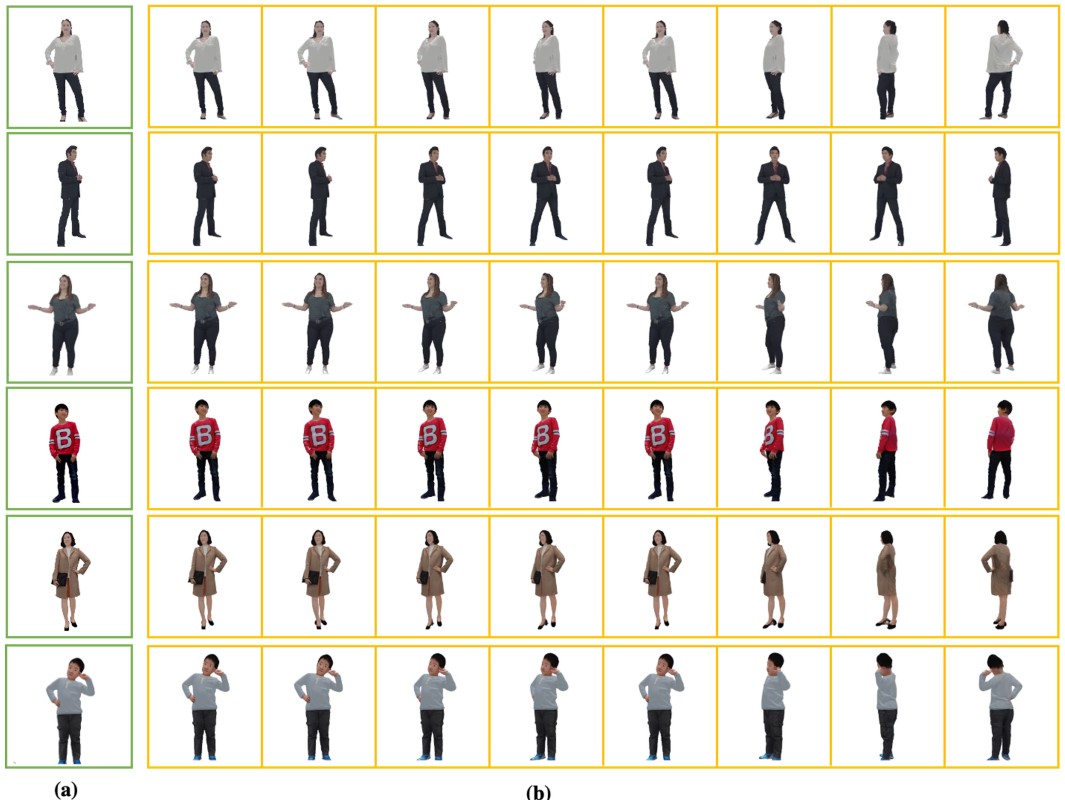

Figure 10: Qualitative 3D Gaussian Splatting results of diversified evaluation dataset. (a) Input Image. (b) Novel view Rendering Results.

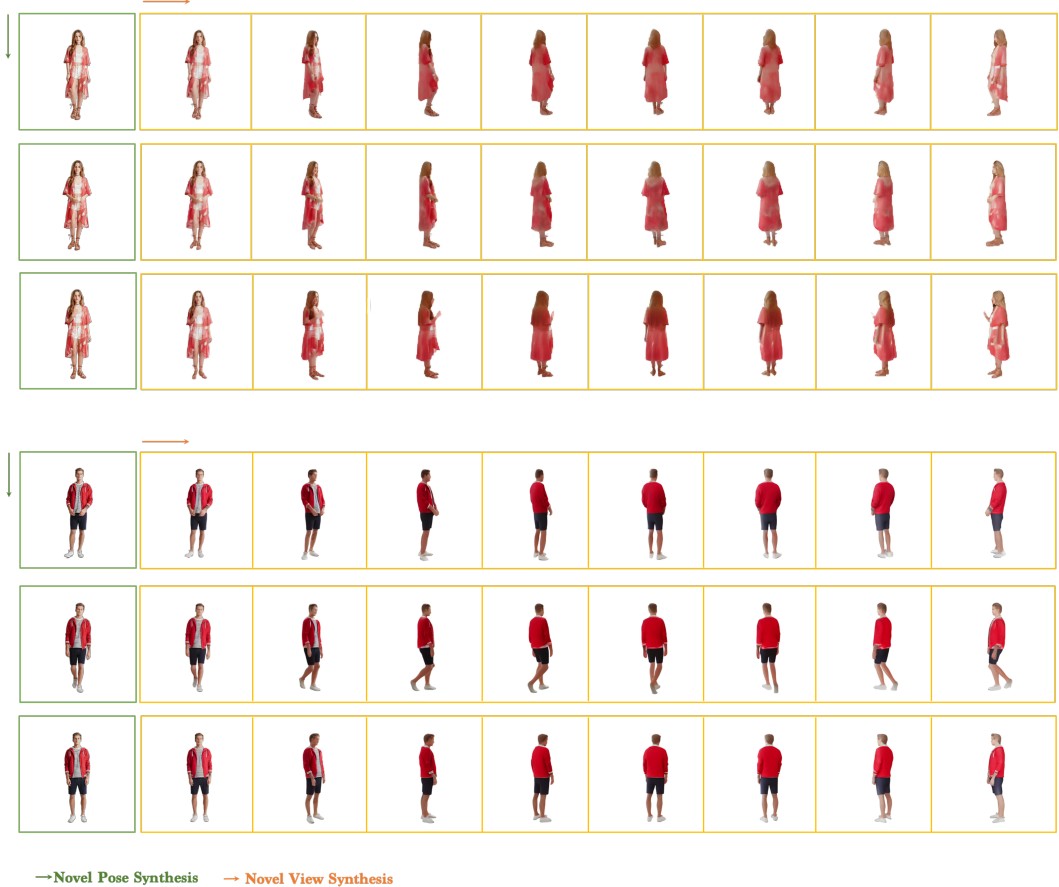

Novel Pose Synthesis    → Novel View Synthesis

Figure 11: Qualitative 4D Gaussian Splatting results on In-the-wild images, including novel view and pose rendering images. (Please zoom in for a detailed view)

