# OpenReview forum: "HumanSplat: Generalizable Single-Image Human Gaussian Splatting with Structure Priors"
_NeurIPS.cc/2024/Conference — NeurIPS 2024 poster_

### Official Review · Reviewer_eC4L · 2024-06-15

**Soundness:** 3
**Presentation:** 2
**Contribution:** 3
**Rating:** 5
**Confidence:** 5

**Summary:**

This paper presents a 3D human reconstruction system that takes a single RGB image and outputs 3D Gaussians, which can render the reconstructed 3D humans to any viewpoint. In contrast to existing 3DGS works, which requires per-instance optimization and cannot be generalized to unseen identities, the proposed system can be used for any unseen identities in zero-shot manner. To achieve this, the authors utilizes 2D mulit-view diffusion model and latent reconstruction transformer with human structure priors. A hierarchical loss, which incorporates human semantic information, is introduced. Experimental results demonstrate the powerful capability of the proposed work.

**Strengths:**

Utilizing diffusion-based generative models to render unseen novel viewpoints is a reasonable choice considering the aiming task is a significantly ill-posed one. The latent reconstruction Transformer effectively incorporates the human structure prior with the generative model’s output. The hierarchical loss could be useful to enforce a consistency between 3D human geometry and appearance.

**Weaknesses:**

1. Unclear writings
Overall, technical details are not enough to fully understand the manuscript.
1-1. For example, Sec. 3.3 describes the video diffusion model. I can’t fully get how the ‘video’ diffusion model can be used for the ‘novel-view’ synthesizer. I understand that a video of an object can model different viewpoints when a camera is moving, but there should be a clear justification and reason why the authors chose this generative model for the novel-view synthesizer. In addition, it seems the video diffusion model does not take the target camera pose. Then which novel view is modeled with the video diffusion model given no camera pose?
1-2. Also, for the hierarchical loss, how can the authors access the target human image of certain body parts? Do the authors already have GT body part segmentation? If the authors simply use rendered body part segmentation from GT SMPL meshes, I don’t think that is a good choice as SMPL meshes only model naked bodies, while there could be hair and loose clothes in the image.

2. Lack of comparison to SOTA methods
Please compare with more SOTA methods, such as ICON and ECON.

3. Lack of in-the-wild results
As the proposed work aims to generalizable 3D human reconstruction, there should be more diverse generalized results. Only three results in Fig. 4 show the results on in-the-wild inputs, which have quite simple poses. Please report more results from input images with diverse **human poses**, identities, and camera pose.

**Questions:**

Please see the weaknesses section.

**Limitations:**

Please see the weaknesses section.

---

> ### Author Rebuttal · Authors · 2024-08-07
>
> We deeply appreciate your recognition of the insight behind our method and its model designs. Below are our clarifications for your concerns.
>
> **Q1:  How the ‘video’ diffusion model can be used for the ‘novel-view’ synthesizer?  Why the authors chose this generative model for the novel-view synthesizer? Which novel view is modeled with the video diffusion model given no camera pose?**
>
> **A1**:
> 1. SV3D [91] is a cutting-edge video diffusion model for **object** novel-view synthesis, and it's finetuned from SVD [93], an image-conditioned video generative model.
> SV3D takes an image and the **relative camera orbit elevation and azimuth angles** as inputs to generate several corresponding novel-view images, which are continuous in 3D space and can be regarded as a video with the camera moving.
> 2. Thanks to the seamless continuity of images in the generated video, SV3D presents state-of-the-art 3D consistency in the object novel-view synthesis task, so we adopt it as the foundational model for our novel-view synthesizer.
> 3. In our work, we finetuned SV3D on human datasets **with relative camera elevation and azimuth angles to generate 4 orthogonal views**, which we found to be sufficient for reconstruction (refer to SIFU [101]).
> 4. We sincerely appreciate the reviewer's keen observation and commit to addressing  **omission of target camera pose input in Equation 1** in the revised manuscript.
>
> **Q2: For the hierarchical loss, how can the authors access the target human image of certain body parts? Do the authors already have GT body part segmentation? While there could be hair and loose clothes in the image, how do the authors deal with it?**
>
> **A2**:
> 1. **Definition.** We employ the widely-used definition of 24 human body parts, such as "rightHand", "leftHand", "head" etc [a]. The GT body part segmentation results are obtained from GT SMPL meshes fitted from scans with predefined semantic vertices and faces. Specifically, we calculate the visible face triangles given the mesh and camera parameters. Each visible triangular face is then assigned the corresponding segmentation label, repeated across all available scans.
>
> 2. **Explanation.**
>
> - **Robustness to hair and clothing**?
> First, **The hierarchical loss is specifically focused on the head and hands**, excluding hair and clothing which can be  further supervised using reconstruction loss $L_{Rec}$.
> Second, instead of using segmentation to reconstruct and then concatenate different components, we design our loss function based on part segmentation. This approach is tolerant of inaccuracies in segmentation.
> - **Comparison.** We also compare our model supervised with a Human Parsing model while still showcasing the current Hierarchical Loss's superiority, as shown in the following table. Besides, compared to 2D image-based human parsing methods, our approach can be utilized online without pre-computation. Additionally, it provides labels with superior 3D consistency, and the supervision signals it offers help to avoid issues associated with training instability.
>
> | |PSNR $\uparrow$| SSIM $\uparrow$  | LPIPS $\downarrow$ |
> | ------------- |:-------------:|:-------------:|:-------------:|
> | with Semantic SMPL (Ours)  |     24.374       |   0.928    |  0.036   |
> | with Human Parsing      |   23.912        |   0.911    |     0.050        |
>
>
> **Q3: Lack of comparison to SOTA methods Please compare with more SOTA methods, such as ICON and ECON.**
>
> **A3**:
> We appreciate the reviewer's suggestion and provide the following detailed explanation:
> 1. Focus on Appearance Results: While ICON (CVPR-2022) and ECON (CVPR-2023) primarily focus on geometry, our work emphasizes appearance results, evaluated through image rendering metrics. These metrics are crucial for assessing final reconstruction fidelity, which geometry-focused methods alone do not capture. Therefore, our comparisons target methods that prioritize both appearance and geometry.
> 2. Comparison to Recent SOTA Methods: We have compared our results with the latest SOTA methods: GTA (NeurIPS 2023), SIFU (CVPR-2024), and TeCH (3DV 2024). These methods have been shown to outperform ICON and ECON in their respective papers. Notably, TeCH is a follow-up work by the same authors of ICON and ECON.
>
> As suggested, we will highlight the rationale for selecting the latest SOTA methods for comparison and cite ECON in the main paper during the revision. Additionally, we will consider including a comparison of our appearance reconstruction results with ICON and ECON's geometry reconstruction results in the supplementary materials.
>
>
> **Q4: Lack of in-the-wild results? As the proposed work aims to generalizable 3D human reconstruction, there should be more diverse generalized results.**
>
> **A4**: We thank the reviewer for the insightful suggestion. In response, we provide more in-the-wild results with diverse human poses, identities, and camera poses in **Figure 2 and Figure 3 of the PDF attachment**  in our "global response" in the rebuttal.
>
>
> ---
> **Reference**：
>
> [a].  https://meshcapade.wiki/assets/SMPL_body_segmentation/smpl/smpl_vert_segmentation.json

---

> > ### Comment · Reviewer_eC4L · 2024-08-11
> >
> > Thanks for the clear answers. Most of the concerns are well addressed.

---

> ### Author Response · Authors · 2024-08-11
>
> **We are pleased to hear that our responses have addressed your major concerns.** We truly appreciate your constructive comments throughout the review process, which have greatly helped in improving our work.

---

> ### Author Response · Authors · 2024-08-13
>
> Dear Reviewer eC4L,
>
> Thank you for your insightful comments and suggestions regarding our manuscript. **We have thoroughly reviewed each point and have made substantial revisions to address the concerns highlighted.**
>
> - Clarified the Novel-view Synthesizer (Sec 3.3) and Hierarchical Loss (Sec 3.5)
> - Compared with HumanSGD on Adobe Stock images.
> - Added qualitative results on in-the-wild images, which reinforce our claims and address your concerns about diverse input images.
>
> If you feel that the rebuttal addresses any of your concerns, **we kindly request that you consider updating your score** in light of these substantial improvements.
> If you have any further questions, **please do not hesitate to contact us.**
>
> We genuinely value the opportunity to enhance our work based on your recommendations.
>
> Best Regards,
>
> Authors of #3255

---

### Official Review · Reviewer_6Egy · 2024-07-07

**Soundness:** 2
**Presentation:** 3
**Contribution:** 2
**Rating:** 5
**Confidence:** 4

**Summary:**

This paper proposes a generalizable human rendering framework from single images. The proposed method relies on different priors and achieves state-of-the-art results on multiple datasets.

**Strengths:**

- The topic of human modeling/rendering from partial observation is important and interesting.
- Thanks to 3DGS, the proposed method achieves a good balance of rendering quality and speed.

**Weaknesses:**

- Qualitative visual results on the rendered images are not satisfactory. It is acceptable that renderings of unseen views of the human can be relatively poorer. It is surprising to see renderings of exactly the input views are also in low quality (Figure 4). This is usually a sign of poor model capacity, which may be attributed to the proposed objective functions or the model itself.
- It is not clear how the 'parts' of human are defined. Are they integer based index as in iuv? If so, how to obtain the ground truth body parts labels? Are they provided by the dataset or they have to be pre-computed by some off-the-shelf models. What if there are prediction errors in the 'ground truth' semantic labels? Is the proposed method robust to this kind of errors? I also have doubts on its effectiveness in terms of the overall rendering quality since there is no ablation study provided to show how it improves quantitative/qualitative results.
- As presented in Table 2 (b), it seems that the proposed method is very sensitive to geometric prior (i.e. SMPL estimations). Since the model relies on PIXIE for SMPL prediction, what if there are errors in the predictions (most likely for challenging poses or complex clothes/background), how will the proposed method perform under those cases? This on the contrary shows advantages of geometric prior free methods such as LRM and HumanLRM. If there is a benchmark on human rendering with challenging poses, I am afraid those methods will perform better than the proposed one.
- I wonder why there are not qualitative comparisons with SIFU but only quantitative metrics?
- I'd suggest to change Figure 1 (b) to a different style. The dashes in the graph are misleading and confuse readers as they are in different scales.

**Questions:**

Please see the weakness section.

**Limitations:**

The authors have adequately addressed the limitations.

---

> ### Author Rebuttal · Authors · 2024-08-07
>
> Thank you very much for your valuable comments! Below are our clarifications for your concerns.
>
> **Q1:  Why are the input views in low quality (e.g., Fig. 4)? Is this a sign of poor model capacity or inappropriate objective functions?**
>
> **A1**:
> 1. (a) HumanSplat reconstructs 3DGS from a single image in a **generalizable** manner, a process that is inherently ill-posed. To enhance its capability to handle diverse scenarios, the quality of the input views is sacrificed for overall balance. For additional details, please refer to **Fig 3 of the attached PDF**. (b) **Similar artifact patterns in the input views** have been reported in Fig. 4 of GTA [30] and Fig. 5 of SIFU [101].
>
> 2. To further confirm the **model's capacity**, we adjusted the weights of the reconstruction loss for the input views ($\times$4), illustrating that the current model with reweighting loss has sufficient capacity to render input views with higher fidelity, as demonstrated in **Fig. 1 of the attached PDF**.
>
>  3. **Ablation on Reweighting Loss Function  on 2K2K  Dataset [104].** While qualitative improvements are noticeable by reweighting loss, there are slight declines in quantitative metrics, showing that reweighting loss is not "a bag of freebies".  We originally chose to balance the quality of the novel view and the input view, and as your suggestion, we will include this analysis in the revision.
>
> |     | PSNR  $\uparrow$ | SSIM  $\uparrow$  | LPIPS  $\downarrow$ |
> | ------------- |:-------------:|:-------------:|:-------------:|
> | w/o reweighting loss (Ours)  |     24.374       |   0.928    |  0.036   |
> | with reweighting loss        |     24.233       |   0.913    |  0.042    |
>
> **Q2: How are human body parts defined? What if there are prediction errors in the 'ground truth' semantic labels? Is the proposed method robust to such errors?**
>
> **A2**:
> 1. **Difinition.** We employ the widely-used definition of 24 human body parts, such as "rightHand", "leftHand", "head" etc. The GT body part segmentation results are obtained from SMPL meshes with predefined semantic vertices [a]. Specifically, we calculate the visible face triangles given the mesh and camera parameters. Each visible triangular face is then assigned the corresponding semantic label. our approach is efficient and can be utilized online without pre-computation. Additionally, it provides labels with superior 3D consistency, and avoid issues associated with training instability.
>
> 2.  **Robustness?**  There are two primary sources of "Hierarchical Loss Error": SMPL misalignment and the influence of hair and clothing. (1) Robustness to SMPL accuracy? During training, we leverage ground truth SMPL parameters to ensure training stability. (2) Robustness to hair and clothing?  To validate this, we compare our model supervised with a Human Parsing model, and the corresponding metrics are as follows:
>
> | |PSNR  $\uparrow$ | SSIM  $\uparrow$  | LPIPS  $\downarrow$  |
> | ------------- |:-------------:|:-------------:|:-------------:|
> | with Semantic SMPL (Ours)  |     24.374       |   0.928    |  0.036   |
> | with Human Parsing      |   23.912        |   0.911    |     0.050        |
>
> The ablation study demonstrates the superiority of our method, primarily because the hierarchical loss is specifically focused on the head and hands, excluding hair and clothing. In contrast, the 2D Human Parsing model does not provide supervision that is consistent in 3D.
>
> **Q3: Is the proposed method very sensitive to geometric priors? How does it handle challenging poses, and what are the advantages over methods that do not use geometric priors?**
>
> **A3**:
> 1. SMPL provides geometric priors that alleviate broken body parts and artifacts (please refer to **Fig. 3 of the attached PDF**). Prior-based methods reduce the demand for large datasets, enhance generalizability, and prevent overfitting. For instance, HumanLRM requires 9.4K scans compared to our method which requires only 3.5K human scans.
>
> 2. We have implemented two key strategies to handle misalignment from imperfect SMPL: (1) Using human priors as keys and values in the Latent Reconstruction Transformer, and (2) Setting a window size $k_{win}$ for fault tolerance. Our generalization ability for challenging poses is shown in **Fig. 3 of the attached PDF**. In the following table, we simulate erroneous poses with random SMPL initialization and demonstrate that our window strategy helps reduce the influence of erroneous human priors.
>
>
> |   | PSNR $\uparrow$| SSIM $\uparrow$  | LPIPS $\downarrow$ |
> | ------------- |:-------------:|:-------------:|:-------------:|
> |  Baseline     |      24.374         |     0.928          |    0.036           |
> |  $k_{win}$=2 + Random SMPL |   19.894      |     0.876    |    0.366   |
> |  $k_{win}$=3 + Random SMPL |    20.609   |    0.883 |    0.327      |
> |  $k_{win}$=4 + Random SMPL |     21.295   |    0.889  |    0.242          |
> |  $k_{win}$=64 (w/o human prior) |    22.635         |    0.893          |    0.182   |
>
>
> 3. For extreme cases, we acknowledge these as limitations. We will add discussion to the limitation part and consider integrating a better SMPL parameter optimization process.
>
> **Q4: Why there are no qualitative comparisons with SIFU?**
>
> **A4**:
> 1. Up to now, the official repository of SIFU [101] only generates coarse geometry and texture. This is inconsistent with their paper because texture refinement significantly impacts the qualitative outcomes. This is unfair to report the qualitative results (**Fig.1 of attached PDF**).
>
> 2. if they update their code during the revision, we promise to complete qualitative results.
>
> **Q5: Change Figure 1 (b) to a different style.**
>
> **A5**:  Thanks for your suggestion, and we will make it more distinct.
>
> We promise to incorporate the addressed issues into the final version.
>
> ---
> **Reference**：
>
> [a]. https://meshcapade.wiki/assets/SMPL_body_segmentation/smpl/smpl_vert_segmentation.json

---

> > ### Comment · Reviewer_6Egy · 2024-08-10
> > **Reply to Rebuttal**
> >
> > I thank the authors for their rebuttal. The additional experiments as well as discussions are insightful. I'd like to raise my rating to borderline accept to acknowledge the efforts the authors put into the rebuttal. However, I still want to note on the poor quality of renderings. As a generalizable methods, it is understandable that unseen regions from the input image are poorly generated/rendered. But it is still not a good sign that the model replicates the input view poorly with blurry and unnatural details --- this may, again, indicate something is wrong with the methodology. The comparison results by HumanSGD, as provided in the rebuttal, however does a much better job replicating the input view.
> > I'd like to acknowledge this paper's strength as well as note the weakness for AC's reference when making the final decision.

---

> ### Author Response · Authors · 2024-08-11
>
> Thank you for your thoughtful review and for taking the time to reconsider our submission after evaluating our rebuttal.
>
> The concern you raised regarding "why HumanSplat cannot replicate the input view in high quality?" is indeed thought-provoking. Upon revisiting our methodology and related works, we recognize that HumanSplat represents 3D content using a set of colored Gaussians ${{\mathcal{G}_i}}$, inherently lacking a direct "shortcut" from the input view to the 3D representation.
>
> Conversely, HumanSGD [22] is equipped with a "Multi-view Fusion" module (please refer to **Figure 5 of HumanSGD**). This module optimizes the UV texture map by minimizing the LPIPS and L1 loss between the input and rendered views. However, the downside is a **time-consuming optimization process** that may lead to random and unstable outcomes, such as the **multi-face Janus issue in novel views.**
>
> **For the unsatisfactory input view's results, we will add a discussion to the limitation.** Notably, compared to generalizable methods such as GTA (NeurIPS 2023), SIFU (CVPR 2024), and LGM (ECCV 2024), HumanSplat exhibits fewer blurry and unnatural details in the input view. Compared to these optimization-based methods (e.g., HumanSGD and TeCH),
> $\textcolor{blue}{\textbf{we offer faster reconstruction times and enhanced robustness.}}$
>
> We deeply value your expertise and time. If you have any further questions, please do not hesitate to contact us.

---

### Official Review · Reviewer_zEoa · 2024-07-13

**Soundness:** 3
**Presentation:** 3
**Contribution:** 2
**Rating:** 6
**Confidence:** 4

**Summary:**

This paper proposes HumanSpat, a method that predicts 3D Gaussians from a single image of a human. The method comprises a 2D multi-view diffusion model and a latent reconstruction transformer that integrates human body prior.

**Strengths:**

+ Well-designed generalizable model that incorporates human prior. The model design is intuitive and clear. The fusion of the human structure prior makes a lot of sense. The end-to-end design addresses major limitations in existing works, such as lacking of human prior makes the prediction missing limbs (as in Human lrm) and two stage approaches where the errors in the human pose prediction affects the clothed reconstruction.

+ Reconstruction time. Powered by 3DGS, the method has minimal reconstruction time and can render novel views at real-time speed.

**Weaknesses:**

- Missing ablation on human prior. The paper should include ablation experiments where there is no human structure prior, as well as experiments where the human structure prior is estimated instead of GT. These experiments can provide insight on the benefit of the specific human structure prior fusion design.

**Questions:**

In Table 1, how was the reconstruction time computed? PIFu is very lightweight and should not take 30 seconds per human.

**Limitations:**

Limitations are addressed.

---

> ### Author Rebuttal · Authors · 2024-08-06
>
> Thank you for your insightful and valuable feedback. It is truly inspiring to know that you appreciate the intuitive and clear design of our model, which offers competitive reconstruction times and can render novel views at real-time speeds.
>
> **Q1: Missing ablation on human prior? The paper should include ablation experiments where there is no human structure prior, as well as experiments where the human structure prior is estimated instead of GT. These experiments can provide insight on the benefit of the specific human structure prior fusion design.**
>
> **A1**: Thank you for your suggestions. Below are our clarifications for your concerns.
> 1. **During training**, we employ SMPL parameters fitted from scans (Ground Truth) to ensure multi-view semantic consistency when computing the hierarchical loss, which helps maintain training stability.
>
> 2. **During testing**, we utilize the off-the-shelf PIXIE [89] to predict SMPL parameters. We report both the estimated and  Ground Truth SMPL parameters and compare the model that does not incorporate a human structure prior, as shown in **Table 2 (b) and (c)  of this paper**. We further demonstrate the results of ablation experiments as follows:
>
> |               | PSNR $\uparrow$| SSIM $\uparrow$  | LPIPS $\downarrow$ |
> | ------------- |:-------------:|:-------------:|:-------------:|
> | GT SMPL parameters| 24.633  | 0.935  | 0.025 |
> | Predicted SMPL parameters|  24.374 (-0.259) | 0.928 (-0.007) |  0.036 (+0.011)|
> | w/o SMPL parameters |  22.635 (-1.998) | 0.893 (-0.0420) | 0.182 (+0.157) |
>
> These experiments provide insight into the benefit of incorporating a human structure prior, as significant performance degradation is observed without it. Furthermore, thanks to our model's fusion design ($k_{win}$), HumanSplat exhibits robustness in testing against challenging poses with imperfect estimation (additional in-the-wild results can be found in **Figure 3 of the supplementary material PDF**), with only minor declines in metrics using predicted SMPL parameters.
>
> **Q2: How was the reconstruction time computed? PIFu is very lightweight and should not take 30 seconds per human.**
>
> **A2**:
> The reconstruction time in Table 1 was calculated from a single image (512$\times$512) to the corresponding 3D representation. Although PiFu is a lightweight network, its inference involves densely sampling the 3D grid and evaluating each point to determine its inclusion in the surface, followed by marching cubes to extract the surface mesh. This runtime and reconstruction resolution are dependent on the number of sampling points. The official implementation of this process takes approximately 30 seconds on two NVIDIA GV100 GPUs, as detailed in the follow-up work [a] by the same authors of PiFu.
>
> ---
> **Reference**:
>
> [a]. Li, Ruilong, et al. "Monocular real-time volumetric performance capture." Computer Vision–ECCV 2020: 16th European Conference, Glasgow, UK, August 23–28, 2020, Proceedings, Part XXIII 16. Springer International Publishing, 2020.

---

> ### Author Response · Authors · 2024-08-13
>
> Dear Reviewer zEoa,
>
> We appreciate your insightful comments and suggestions on our manuscript. We have thoroughly reviewed each point to address the highlighted concerns:
>
> - Detailed the ablation experiments both without human structure priors and with **estimated human structure priors** (PIXIE).
> - Explained the PIFu's time-consuming procedure.
>
> If you find the rebuttal addresses your concerns or have any further questions, please feel free to contact us.
>
> We truly value the chance to refine our work based on your invaluable feedback.
>
> Best Regards,
>
> Authors of #3255

---

### Official Review · Reviewer_2EjT · 2024-07-18

**Soundness:** 3
**Presentation:** 3
**Contribution:** 3
**Rating:** 6
**Confidence:** 4

**Summary:**

- The paper, HumanSplat, focuses on photorealistic novel-view synthesis of humans from a single image.
- The key idea is to use a multi-view synthesizer based on SV3D to hallucinate the other views + latent reconstruction transformer to predict the 3DGS.
- Importantly, HumanSplat does not use optimization and is much faster than existing methods.
- Baselines:  Magic, HumanSGD, TeCH
- Evaluations are done on Thuman2.0 and Twindom datasets. Metrics: PSNR, SSIM, LPIPS
- The proposed methods consistently outperforms the existing methods.

**Strengths:**

- The paper is well-written, organized and easy to follow.
- The key idea of multi-view diffusion plus transformer to predict parameters of gaussians in the context of human avatar creation is novel.
- An important technical contribution is the archictecture of the latent reconstruction transformer with geometry-aware interaction.
- The experiments are done on multiple datasets along with informative ablative studies.
- Table. 1 highlights the speed-up gains compared to existing methods. HumanSplat is almost as fast as LRM but utilizes an explict human prior and achieves much better performance.

**Weaknesses:**

- Measuring 3D consistency of SV3D: The novel views generated by fine-tuned SV3D is important for predicting the gaussian parameters. An ablative study on the effectiveness of the novel synthesizer would be really helpful to understand the in-the-wild generalization of the proposed method. 3D consistency could be measure using stereo-based matching or local correspondences.

- Qualitative results: Although, Table. 1 showcases better performance than HumanSGD on THuman2.0 dataset, notably in all metrics. The qualitative comparison between HumanSGD and HumanSplat is missing. HumanSGD results demonstrate much better modelling of human skin tone and overall 3D geometry. I understand the code is yet to be released, however, would it be possible to compare on similar stock images?

- Minor: consider renaming "project-aware" -> "projection-aware". The writing at times is a little too non-technical, eg. L206-207, consider rephrasing such instances.

**Questions:**

- Understanding the generalizability of novel-view synthesizer. Is it limited to full-body images? Any quantitative insights into 3D consistency.
- Qualitative comparisons to HumanSGD on stock images.

**Limitations:**

Yes. The authors provide a discussion about utilizing 2D data and handling diverse clothing styles.

---

> ### Author Rebuttal · Authors · 2024-08-06
>
> We thank the reviewer for the insightful and thorough feedback. It is inspiring to hear that you find the paper well-written, organized, and easy to follow, that the method in the context of human avatar creation is novel, and that our experiments are comprehensive. Below are our clarifications for your concerns.
>
> **Q1: Measuring the 3D consistency of novel views generated by fine-tuned SV3D and providing any quantitative insights into 3D consistency?**
>
> **A1**: Your insight regarding the novel-view synthesizer's 3D consistency metrics is enlightening.  As suggested, we utilize the cutting-edge local correspondence matching algorithm, MASt3R [a], to measure the **3D consistency of the novel-view synthesizer**.  Specifically, we performed one-to-one image matching between input views and their generated novel views, using the average number of matching correspondences as a metric.
>
> Quantitatively compared to the original SV3D [93], our synthesizer demonstrates a significant enhancement in 3D consistency for human novel-view generation after fine-tuning using human datasets.
>
> |                 | Matching Points Number |
> | -------------   |:-------------:|
> |   SV3D [93]   |   723.25          |
> |  Novel-view Synthesizer (Ours) |  930.13           |
> |  Ground Truth   |   1106.33         |
>
> We will incorporate it into the ablation study that fine-tuning on human datasets improves the multi-view consistency metrics.
>
> **Q2: The generalizability of the novel-view synthesizer (including in-the-wild scenarios)? Is it limited to full-body images?**
>
> **A2**: The novel-view synthesizer, inheriting the generalization and multi-view consistency of SV3D [93] and fine-tuned on human datasets, exhibits robust multi-view generation capabilities and has enhanced generalization abilities within the human domain for different regions and in-the-wild scenarios, including 'head only,' 'upper body,' and 'full body.' Our SV3D model outputs intermediate feature maps, which are not visually indicative of quality. However, the human GS generation based on in-the-wild images demonstrates the generalization ability of our synthesizer, as shown in **Figure 2 and Figure 3 of the PDF attachment**.
>
> **Q3: The qualitative comparison between HumanSGD and HumanSplat is missing. How does it compare with HumanSGD in "in-the-wild" images from the Adobe Stock Website?**
>
> **A3**: In response to your insightful request and suggestion, we provided a qualitative comparison against HumanSGD in-the-wild image from the Adobe Stock Website in **Figure 2 of the PDF attachment**. Notably, our results demonstrate satisfactory outcomes without per-instance optimization.
>
> **Q4: Minor issues.**
>
> **A4**: We will rename it to "projection-aware" and modify L206-207 to "Therefore, to enhance the fidelity of reconstructed human models, we optimize our objective functions to preferentially focus on the facial regions." in our revision, eliminating non-technical parts and ensuring consistency throughout the manuscript.
>
>
> ---
> **Reference**：
>
> [a]. Leroy, Vincent, Yohann Cabon, and Jérôme Revaud. "Grounding Image Matching in 3D with MASt3R." arXiv preprint arXiv:2406.09756 (2024).

---

> ### Author Response · Authors · 2024-08-13
>
> Dear Reviewer 2EjT,
>
> We appreciate your insightful comments and suggestions on our manuscript. We have thoroughly reviewed each point to address the highlighted concerns:
>
> - Measured the 3D consistency metric of novel-views synthesizer based on your recommendation.
> - Conducted comparisons with HumanSGD on Adobe Stock images.
> - Addressed and clarified minor issues highlighted in your feedback.
>
> If you find the rebuttal addresses your concerns or have any further questions, please feel free to contact us.
>
> We truly value the chance to refine our work based on your invaluable feedback.
>
> Best Regards,
>
> Authors of #3255

---

> > ### Comment · Reviewer_2EjT · 2024-08-13
> >
> > Thank you for the rebuttal. My concerns are addressed, I would like to keep the rating of weak accept (6).

---

> > > ### Author Response · Authors · 2024-08-13
> > >
> > > **We are pleased to hear that our responses have addressed your concerns.** We truly appreciate your constructive comments throughout the review process, which have greatly helped in improving our work.

---

### Author Rebuttal · Authors · 2024-08-07

We would like to express our gratitude to all the reviewers for their valuable, constructive, and thoughtful feedback. It is truly inspiring to hear that the majority of reviewers recognize that:
- The proposed method is meaningful (6Egy), efficient (2EjT, 6Egy), and addresses major limitations in existing work (zEoa).
- The quantitative and qualitative experiments are extensive (2EjT).
- The model design is novel and make sense (2EjT, zEoa, eC4L).
- The paper is well-written, well-organized, and easy to follow (2EjT).

We have responded thoroughly to all reviewers **in the corresponding rebuttal text input box** and have included additional qualitative results in the **supplementary material PDF** to address concerns raised by the reviewers.

**Answer for unsatisfactory results in input views**:
In Figure 1, we adjusted the weights of the reconstruction loss $L_{Rec}$ between the input and novel views, resulting in improved quality of the input views. Simultaneously, we showcased the qualitative results of SIFU without  texture refinement.




**Answer for in-the-wild results**:
In Figure 2, we provided additional results for in-the-wild images sourced from **Adobe Stock**. In Figure 3, we demonstrate the generalizability of our model with in-the-wild images in **challenging scenarios**, and highlight its superiority over Geometric Prior Free methods (e.g., LGM) in alleviating broken body parts and artifacts.

Thank you very much for your time and valuable input! If there are any further concerns that we have not yet addressed, we would be eager to hear from you.

---

### Decision · Program_Chairs · 2024-09-25

**Decision:**

Accept (poster)

**Comment:**

- This paper presents a model for single-view 3D human reconstruction with Gaussian splats. It uses a fine-tuned diffusion-based novel view synthesizer to generate multiple views of the input and proposes a prior-aware latent reconstruction model for Gaussian splat prediction. The reviewers (**2EjT,**  **zEoa, eC4L**) ****acknowledge the technical contribution of using masked cross-attention to incorporate the SMPL mesh prior as tokens.
- Concerns are mainly raised around the influence of introduced SMPL prior (**zEoa, 6Egy,  eC4L**) and missed visual comparisons (**2EjT, 6Egy)**
- In the rebuttal, the authors did a good job of addressing the questions by providing ablation for SMPL pose priors and additional visual results for HumanSGD and in-the-wild results.
- The AC agrees with the reviewers and recommends its acceptance, and also requests the authors to incorporate the reasoning and results in the responses to the final version.